Land use scenarios, seasonality, and stream identity determine the water physicochemistry of tropical cloud forest streams

Vázquez Gabriela 1
http://orcid.org/0000-0001-9985-5719 Ramírez Alonso 2 alonso.ramirez@ncsu.edu
http://orcid.org/0000-0002-2737-9327 Favila Mario E. 1
http://orcid.org/0000-0003-0098-0806 Alvarado-Barrientos M. Susana 1
1 Instituto de Ecología, A.C. , Xalapa, Veracruz , Mexico
2 Department of Applied Ecology, North Carolina State University , Raleigh, NC , USA
Christoffersen Kirsten
Electronic publication date: 2023 Jun 5
Publication date: 2023
Volume: 11
Electronic Location ID: e15487
Received 2022 Sep 16; Accepted 2023 May 10
Copyright: © 2023 Vázquez et al.
Copyright year: 2023
Copyright holder: Vázquez et al.
License: This is an open access article distributed under the terms of the Creative Commons Attribution License, which permits unrestricted use, distribution, reproduction and adaptation in any medium and for any purpose provided that it is properly attributed. For attribution, the original author(s), title, publication source (PeerJ) and either DOI or URL of the article must be cited.
License URL: https://creativecommons.org/licenses/by/4.0/

Keywords: Tropical streams, Land use, Cloud forest, Stream physicochemistry, Land use scenarios, Seasonality

Funding: Mexican Consejo Nacional de Ciencia y Tecnología (CONACyT) 285962 This work was supported by the Mexican Consejo Nacional de Ciencia y Tecnología (CONACyT), Grant No. 285962. The funders had no role in study design, data collection and analysis, decision to publish, or preparation of the manuscript.

==============================
Background

Land use is a major factor determining stream water physicochemistry. However, most streams move from one land use type to another as they drain their watersheds. Here, we studied three land use scenarios in a tropical cloud forest zone in Mexico. We addressed three main goals, to: (1) assess how land use scenarios generate different patterns in stream physicochemical characteristics; (2) explore how seasonality (i.e., dry, dry-to-wet transition, and wet seasons) might result in changes to those patterns over the year; and (3) explore whether physicochemical patterns in different scenarios resulted in effects on biotic components (e.g., algal biomass).

Methods

We studied Tropical Mountain Cloud Forest streams in La Antigua watershed, Mexico. Streams drained different three scenarios, streams with (1) an upstream section draining forest followed by a pasture section (F-P), (2) an upstream section in pasture followed by a forest section (P-F), and (3) an upstream forest section followed by coffee plantation (F-C). Physicochemistry was determined at the upstream and downstream sections, and at the boundary between land uses. Measurements were seasonal, including temperature, dissolved oxygen, conductivity, and pH. Water was analyzed for suspended solids, alkalinity, silica, chloride, sulfate, magnesium, sodium, and potassium. Nutrients included ammonium, nitrate, and phosphorus. We measured benthic and suspended organic matter and chlorophyll.

Results

Streams presented strong seasonality, with the highest discharge and suspended solids during the wet season. Scenarios and streams within each scenario had distinct physicochemical signatures. All three streams within each scenario clustered together in ordination space and remained close to each other during all seasons. There were significant scenario-season interactions on conductivity (F = 9.5, P < 0.001), discharge (F = 56.7, P < 0.001), pH (F = 4.5, P = 0.011), Cl− (F = 12.2, P < 0.001), SO42− (F = 8.8, P < 0.001) and NH4+ (F = 5.4, P = 0.005). Patterns within individual scenarios were associated with stream identity instead of land use. Both P-F and F-C scenarios had significantly different physicochemical patterns from those in F-P in all seasons (Procrustes analysis, m12 = 0.05–0.25; R = 0.86–0.97; P < 0.05). Chlorophyll was significantly different among scenarios and seasons (F = 5.36, P = 0.015, F = 3.81, P = 0.42, respectively). Concentrations were related to physicochemical variables more strongly during the transition season.

Conclusion

Overall, land use scenarios resulted in distinctive water physicochemical signatures highlighting the complex effects that anthropogenic activities have on tropical cloud forest streams. Studies assessing the effect of land use on tropical streams will benefit from assessing scenarios, rather than focusing on individual land use types. We also found evidence of the importance that forest fragments play in maintaining or restoring stream water physicochemistry.

Introduction

Land use activities and soil type characteristics on the watershed determine many of the physicochemical characteristics of stream ecosystems (Allan, 2004; Poff, Bledsoe & Cuhaciyan, 2006). Agricultural areas, for example, are major sources of nutrients and pesticides to streams (Chase et al., 2016). At large spatial scales (e.g., regions), the physical and chemical characteristics of stream ecosystems can be predicted based on land use patterns (Burcher, Valett & Benfield, 2007; Maloney & Weller, 2011). At small scales (stream reaches), riparian areas are points of interaction between the terrestrial and aquatic environments and play a role determining stream characteristics (Chase et al., 2016). Deforestation of riparian forest, even in small scales, produces major increments of temperature and sediment loads (Herlihy, Stoddard & Johnson, 1998; Chase et al., 2016). Understanding the effects of land use on stream physicochemical characteristics, help us understand factors that could impact the entire ecosystem, like changes in basal resource availability or the composition of biotic communities (Benstead, Douglas & Pringle, 2003; Lorion & Kennedy, 2009).

Streams often drain more than a single land use type and watersheds could be best visualized as mosaics of a variety of land uses patches (Allan, 2004). Those mosaics create sequences of different land uses along the stream continuum, adding complexity to the interaction between land use and stream responses (Johnson et al., 1997). This is particularly true in many tropical areas, where agricultural practices occur at small scales (Ríos-Touma & Ramírez, 2018; Nunes et al., 2022). In tropical mountains, streams often drain from forest patches into agricultural areas, returning to forest in some cases (Encalada et al., 2019). While our knowledge of stream ecosystem responses to individual land uses is extensive (e.g., Allan, 2004), we have limited information on the consequences of different land use sequences on water physicochemistry or stream ecosystem dynamics.

In the tropical cloud forest of Mexico, our study area, streams drain a patchwork of land uses composed mainly of primary and secondary forests, pastures for cattle grazing, and plantations of shaded coffee. Most of these activities are conducted at small spatial scales (few hectares) and streams often drain multiple land uses. However, our understanding of the effects of land use on cloud forest streams comes from studies focused on stream reaches draining single land uses (Vázquez, Aké-Castillo & Favila, 2011). Four dominant scenarios that are frequently found in this area are: streams that start within forest and drain into pasture or coffee plantation, and their opposite combinations of streams draining first pasture and coffee plantation that move into forest fragments. As in many other tropical regions, we lack an understanding of how these combinations of land use patches create scenarios that affect stream physicochemistry. We expect those scenarios to result in different physicochemical characteristics and therefore to inform watershed management and conservation.

The magnitude of land use effects on stream physicochemistry changes seasonally. In tropical regions, where dry and wet seasons can be delimited, the strongest effects over stream ecosystems often occur during the wet season when runoff increase substantially (Muñoz-Villers & McDonnell, 2012). Increased precipitation and runoff facilitate the movement of solutes from terrestrial to aquatic ecosystems, not only through surface flow along slopes, but also as in-stream bank erosion. During the wet season, streams receive increased amounts of suspended particles and water solutes, while biotic components, like algal biomass, decrease (Vázquez, Aké-Castillo & Favila, 2011). In contrast, the dry season is dominated by base flow and stream physicochemistry could be expected to reflect within channel processes, as nutrient recycling and benthic metabolism. Biotic components (e.g., algal biomass), increase during base flow conditions due to reduced flood disturbance.

Here, we assessed water physicochemical characteristics under different land use scenarios (forest-pasture, forest-coffee plantation, pasture-forest) in streams draining tropical cloud forest in Mexico. Our main goal was to assess how land use scenarios generate particular patterns in stream physicochemical characteristics. In addition, we explored how seasonality (i.e., dry, transition to wet, and wet season) might change those patterns over the year. Finally, we explored whether physicochemical patterns in different scenarios resulted in effects on biotic components (e.g., algal biomass).

Materials and Methods

Study area and sampling sites

Our study area is located in the headwaters of the La Antigua watershed, a major watershed that drains part of the states of Puebla and Veracruz before discharging into the Gulf of Mexico. We selected streams located within the Tropical Mountain Cloud Forest (TMCF), at elevations between 1,100 and 2,200 m a.s.l. The climate of this zone is temperate humid and 80% of annual precipitation falls as convective storms during the wet season (May/June–October). During the dry season (November–April) the weather is mostly stable and dry, but eventual cold fronts produce light rains and fog. Mean annual rainfall in this altitudinal strip ranges from 1,385 mm to 3,185 mm and mean daily temperature is 15 °C to 19 °C (Muñoz-Villers & McDonnell, 2013). The soils of the entire study area are Andosols formed on shallow volcanic ash deposits over semi-permeable fractured andesitic-basaltic bedrock (Rossignol et al., 1987).

TMCF is one of the most biodiverse forests in Mexico (Rzedowski, 1996; Ruiz-Jiménez, Téllez-Valdés & Luna-Vega, 2012). The area is under severe deforestation pressure and land use conversion from forest into different agricultural practices and urbanization is common (Espinoza-Guzmán, Borrego & Sahagún-Sánchez, 2023). Current land use in the area is a mosaic of forest (mostly restricted to steep slopes), agricultural fields, shade coffee plantations, pastures, and urban areas (Hernández-Pérez et al., 2022; Martínez et al., 2009). Agricultural land uses typically continue along the slopes down to the streambank without a differentiated riparian vegetation buffer, rather, sometimes only dispersed remnant TMCF vegetation occurs a long a narrow riparian strip (<5 m wide).

Nine perennial headwater streams were selected for the study (Fig. 1). In each stream, we identified the transition between two contrasting land uses (e.g., forest, pasture, coffee plantation) and selected a 100 m section in each land use (for a combined 200 m study reach per stream) to set up three scenarios (see also Table 1 and Fig. 1). Scenario F-P = streams with upstream forest followed by pasture; Scenario P-F = streams with upstream pasture followed by forest; and Scenario F-C: streams with upstream forest followed by shaded coffee plantation. The fourth possible scenario, streams that drain coffee plantation and move into forest, was not included due to lack of accessible sites under this scenario. Forest stream sections had dense riparian vegetation of either natural forest or secondary growth. The riparian zone of shaded coffee plantation sections included the same shade tree species as the plantations typically reach the stream margin. Pasture were for cattle grazing and some had a narrow strip of riparian vegetation present (i.e., scattered trees and shrubs remnant of the former forest).

Figure 1 Study streams draining different scenarios in the upper La Antigua watershed, Veracruz, Mexico.

The nine study streams were first or second order draining Forest-Pasture (F-P, green stars), Pasture-Forest (P-F, yellow circles), and Forest-Coffee plantation (F-C, red pentagons). Since we selected first to second order streams, not all symbols fall on a visible stream channel.

Table 1 Study stream characteristics per scenario at La Antigua Watershed, Mexico.

			Elevation	Slope	Width	Canopy cover (%)	Slope riparian	Latitude	Longitude	
	Stream	Order	(m a.s.l.)	(%)	(m)	Upstream	Downstream	zone (degrees)	(°N)	(°W)	
Forest—Pasture scenario									
	1	2	1,869	8	4.2	92	43	27	19.4058	97.0978	
	2	1	1,730	8	3.9	73	43	45	19.4052	97.0819	
	3	1	1,421	5	1.8	82	78	30	19.5287	96.9763	
Pasture—Forest scenario									
	4	1	1,530	10	2.9	89	90	43	19.3654	97.0580	
	5	2	1,780	9	5.4	44	81	33	19.4027	97.0576	
	6	1	1,420	3	1.4	19	87	32	19.3883	97.0493	
Forest—Coffee scenario									
	7	1	1,070	6	2.5	80	63	31	19.3527	96.9724	
	8	1	1,285	12	2	73	68	33	19.4635	96.9924	
	9	1	1,420	18	3.9	83	78	52	19.4702	96.9892	

Data collection

Mean canopy cover over the stream was estimated with the HabitApp mobile application by averaging leaf coverage of 10 photographs taken with a leveled cell phone at points separated ~10 m along each study section. The channel and riparian zone slopes were measured using a clinometer. Physicochemical characterization of stream water was conducted with samples collected at three points along each 200 m-long reach under study: at the upper most point (0 m), a middle point which is the boundary between the two land uses (100 m), and at the end of downstream section (200 m). Sampling was conducted during the dry (March–April), dry-to-wet transition (June), and wet (October) seasons of 2019.

We estimated discharge at each sampling point by measuring depth and instantaneous velocity at 10 cm intervals across a perpendicular transect. Flow was measured with a Flow Probe 101-FP201 by placing the sensor at 60% of depth at each sampling point. Discharge (m3/s) was calculated as Q = Av, where A is the transversal area and v is flow (Gore & Banning, 2017). Water temperature (°C), dissolved oxygen (mg/L), and conductivity (μS/cm) were measured with a portable Yellow Spring Instruments meter (YSI, Model 85), and pH with a Barnant potentiometer (Model 20). A water sample was collected in polyethylene bottles (1 L) and filtered using Whatman GF/C filters to determine physical and chemical variables: total suspended solids (TSS, dry weight), alkalinity (as CaCO3, phenolphthalein), silica (SiO2, molybdate), chloride (Cl−, titration with phenolphthalein) and sulfate (SO42−, turbidimetric technique). Calcium (Ca2+) and magnesium (Mg2+) were measured using an atomic absorption spectrophotometer (Shimadzu Mod. AA6501), and sodium (Na+) and potassium (K+) with a flame photometer (Corning Mod. 410). Nitrogen was measured as ammonium (N-NH4+, Nessler), nitrate (N-NO3−, brucine), we also report dissolved inorganic nitrogen (DIN) as the sum of those two components. A second filtered water sample was collected using 250-ml glass bottles to measured reactive phosphorus (PO4, ascorbic acid) and another one unfiltered to determine total phosphorus (TP, persulfate digestion, and ascorbic acid). Determinations were conducted per APHA spectrophotometric techniques (APHA, 2005). All samples were kept refrigerated (4 °C) until analysis.

Benthic organic matter (BOM) was estimated by collecting three samples with a corer sampler (14 cm diameter) in areas of pools at each study section (six per stream) following methods in Lamberti et al. (2017). Coarse matter was manually collected using a 1 mm sieve. Then, sediments within the corer were disturbed and the water filtered through 250 and 65 µm sieves. Up to 5 L of water were filtered for fine particles. All samples were transported to the laboratory and dried for 24 h at 70 °C to obtain dry mass and ashed for 1 h at 500 °C to obtain ash free dry mass (AFDM). Both fractions were combined to obtain BOM. Suspended organic matter (SOM) was determined by filtering the water column using a drift net (mesh 250 μm) for 15–20 min. We measured the area of net submerged and water flow to obtain the volume sampled. For fine SOM, 20 L of water were filtered using a 65 μm sieve. Samples were processed as for BOM and both fractions were added to obtain SOM. Chlorophyll a was assessed by collecting three rocks (upper side of ~10 cm2) per sampling point and placing them in separate bottles with 90% methanol. Samples were kept cool and in the dark until analyzed within 24 h in the laboratory. Chlorophyll a was measured spectrophotometrically and concentration (mg/m2) was determined through Holden’s equations (Meeks, 1974). We obtained rock area using aluminum foil to cover the area exposed to solar radiation (i.e., upper side of rock) and applied a weight-area conversion.

Statistical analyses

Principal components analysis (PCA) was applied to the correlation matrix to identify patterns in water physicochemistry between the different scenarios and seasons. The variables were transformed with log10 (X+1) before the analysis.

A two-way ANOVA was applied to evaluate how the scenario, season, and their interaction affected each physicochemical variable measured. We used a Shapiro test of normality and a Fligner-Killeen test of homogeneity of variances in each case. When normality or homogeneity of variance were not met, we used a Kruskal Wallis test. Streams were nested within scenarios. Tests were run using the statistical package R, version 4.1.2 (R Core Team, 2021).

We applied a PCA to each scenario independently to assess within scenario patterns during each season. A Procrustes analysis was performed to assess whether the three land use scenarios resulted in different stream physicochemistry during each season (dry, transition, wet). Procrustes analysis rotates and scales an ordination (a PCA in our case) to allow for the comparison of two ordinations and provides a correlation coefficient and P-value between the first axes of the two ordinations. We performed two comparisons per season: F-P against P-F and F-C. We considered F-P as the scenario with the potential lowest impact, as the upstream section drains forest while the downstream is used as pasture, thus we used this scenario as the reference. We generated a PCA for each scenario using three streams and three sampling points per stream: upstream (forest or pasture), middle point, and downstream (forest, pasture, or coffee). Analyses were run in R using the package Vegan and the functions Procrustes and Protest with 9,999 permutations.

To assess the effects of physicochemistry on biotic components (i.e., chlorophyll concentration), we used Pearson correlations to relate chlorophyll concentrations against major physicochemical variables at each scenario.

Results

Stream physicochemistry

Our cloud forest streams presented strong seasonality, with the highest discharge values (up to 248.52 m3/s) during the wet season (Table 2). It is noteworthy that the discharge on the F-C on the wet season was almost as low as the other scenarios in the dry season. Water temperatures ranged from 14.63 to 19.66 °C and had moderately oxygenated water (range: 5.96 to 6.55 mg/L). Water pH was slightly acidic to neutral in most streams, range 5.52 to 6.96, with lowest values recorded during the wet season (Table 2). High TSS concentrations were observed in the wet season, with the highest values (14 mg/L) present in streams draining coffee plantation (Table 2).

Table 2 Water physicochemistry per scenario and season.

Values are means (n = 3) for each scenario and season with standard error in parenthesis.

	1-Forest—Pasture		2-Pasture—Forest		3- Forest—Coffee plantation	
	Dry	Transition	Wet		Dry	Transition	Wet		Dry	Transition	Wet	
Discharge (m3/s)	20.26	47.32	248.52		28.99	66.41	213.21		8.57	15.98	21.20	
	(5.43)	(3.70)	(8.79)		(0.79)	(18.49)	(9.97)		(2.17)	(7.18)	(1.27)	
Temperature (°C)	14.63	15.86	15.63		15.18	17.12	16.97		16.64	19.66	18.42	
	(0.04)	(0.02)	(0.02)		(0.44)	(0.11)	(0.13)		(0.07)	(0.15)	(0.05)	
Oxygen (mg/L)	6.39	6.55	6.52		5.96	6.03	6.04		6.10	6.55	6.17	
	(0.04)	(0.01)	(0.01)		(0.18)	(0.08)	(0.19)		(0.05)	(0.15)	(0.03)	
pH	6.93	6.96	6.50		6.62	6.59	5.52		6.72	6.52	6.27	
	(0.07)	(0.03)	(0.00)		(0.06)	(0.18)	(0.01)		(0.12)	(0.20)	(0.14)	
Conductivity (µS/cm)	32.63	30.29	25.15		23.64	26.96	20.96		49.52	49.81	41.63	
	(0.70)	(0.52)	(0.64)		(1.15)	(0.33)	(0.15)		(0.43)	(0.32)	(0.27)	
TSS (mg/L)	4.64	11.67	10.35		11.06	8.81	13.33		9.12	10.00	14.40	
	(1.81)	(2.88)	(1.99)		(3.33)	(1.89)	(4.10)		(2.28)	(2.63)	(6.82)	
Alkalinity (mg CaCO3/L)	23.38	21.70	16.09		19.18	18.89	16.13		31.92	28.41	14.13	
	(0.14)	(0.31)	(0.23)		(0.35)	(0.07)	(0.12)		(0.09)	(0.05)	(0.79)	
N-NH4 (µM)	3.11	5.81	4.20		0.31	6.85	1.30		4.19	7.22	8.38	
	(0.34)	(0.34)	(1.28)		(0.26)	(1.47)	(1.31)		(0.28)	(0.82)	(0.53)	
N-NO3 (µM)	14.88	16.65	18.31		11.04	12.05	9.62		16.26	15.17	29.70	
	(0.31)	(0.52)	(0.59)		(0.88)	(1.56)	(0.32)		(0.75)	(0.39)	(0.40)	
DIN (µM)	18.00	22.46	22.70		11.39	18.91	10.93		20.45	435.93	38.07	
	(0.09)	(0.31)	(0.80)		(1.11)	(2.96)	(0.31)		(0.51)	(221.77)	(0.72)	
PO4 (mg/L)	0.47	0.36	0.42		0.33	0.25	0.19		0.71	0.33	0.59	
	(0.10)	(0.12)	(0.06)		(0.06)	(0.01)	(0.01)		(0.20)	(0.04)	(0.06)	
TP (mg/L)	0.47	0.36	0.42		0.33	0.25	0.19		1.33	1.04	1.02	
	(0.10)	(0.12)	(0.06)		(0.06)	(0.01)	(0.01)		(0.13)	(0.29)	(0.16)	
DIN/PT	20.77	28.84	34.82		13.46	26.36	30.66		15.68	25.29	39.14	
	(1.70)	(2.21)	(4.93)		(1.20)	(5.91)	(1.21)		(1.82)	(6.89)	(5.81)	
SiO2 (mg/L)	26.31	22.16	20.45		18.86	15.89	14.52		36.33	28.27	28.77	
	(0.07)	(0.09)	(0.13)		(0.10)	(0.20)	(0.10)		(0.30)	(1.43)	(0.15)	
Cl− (mg/L)	5.50	7.00	8.56		5.78	4.98	7.75		6.37	5.64	9.41	
	(0.11)	(0.24)	(0.27)		(0.38)	(0.04)	(0.05)		(0.21)	(0.23)	(0.15)	
SO42− (mg/L)	0.62	1.01	1.50		0.36	1.05	1.16		0.65	0.86	1.86	
	(0.11)	(0.07)	(0.07)		(0.03)	(0.06)	(0.07)		(0.11)	(0.05)	(0.07)	
Ca2+ (mg/L)	1.10	1.23	0.85		1.38	0.40	1.15		1.01	0.75	0.90	
	(0.02)	(0.28)	(0.04)		(0.14)	(0.18)	(0.12)		(0.01)	(0.03)	(0.05)	
Na+ (mg/L)	5.80	3.31	2.91		4.66	1.37	2.47		6.85	9.03	8.60	
	(1.91)	(0.12)	(0.31)		(0.42)	(0.05)	(0.51)		(0.11)	(0.44)	(1.36)	
Mg+ (mg/L)	1.47	1.17	0.53		1.11	0.88	0.38		1.56	1.30	1.25	
	(0.32)	(0.13)	(0.03)		(0.23)	(0.03)	(0.07)		(0.06)	(0.03)	(0.10)	
K+ (mg/L)	0.35	0.03	0.00		0.46	0.00	0.00		1.33	1.30	0.70	
	(0.00)	(0.02)	(0.00)		(0.08)	(0.00)	(0.00)		(0.04)	(0.05)	(0.08)	
BOM (g/m2)	5,227.58	3,044.16	1,020.52		4,203.02	4,171.88	1,081.59		3,732.17	1,852.90	1,130.62	
	(1,344.12)	(349.14)	(121.16)		(531.48)	(465.74)	(199.64)		(1,647.16)	(269.41)	(131.09)	
SOM (mg/L)	607.98	1,003.70	45,389.43		770.57	1,641.49	5,143.52		390.89	425.92	556.81	
	(114.24)	(154.00)	(38,716.13)		(47.30)	(503.99)	(391.96)		(41.66)	(219.15)	(69.93)	
Chlorophyll a (mg/m2)	1.20	0.47	0.25		0.71	0.27	0.78		0.91	0.90	0.40	
	(0.59)	(0.07)	(0.03)		(0.23)	(0.09)	(0.13)		(0.15)	(0.07)	(0.05)	

Alkalinity was similar among streams, range 16.09 to 31.92 mg CaCO3/L, but the highest values were recorded in scenarios that included coffee plantation. Alkalinity decreased during the wet season in all scenarios (Table 2). Most streams were nutrient poor, except for those draining coffee plantation (Table 2). For example, TP ranged from 0.19 to 0.47 µM in streams draining forest and pasture, but those that drained coffee plantation had over two times those concentrations (range: 1.02 to 1.33 µM, Table 2). Most ions showed a tendency to decrease in concentration during the wet season, except for Cl− and SO42− that increased (Table 2).

BOM ranged from 1,020 to 5,227 g/m2, with both extreme values recorded in the forest-pasture scenario (Table 2). SOM ranged from 390 mg/L in F-C during the dry season to 45,389 mg/L in F-P during the wet season. Wet season values in the F-C scenario were highly variable due to a single sample that measured twice the amount of SOM than the rest (Table 2).

PCA analysis generated groups that corresponded to our scenarios and seasons (Fig. 2). Axis 1 accounted for 36% of the variance, with axis 2 accounting for an additional 28%. Positively associated with Axis 1 were alkalinity, SiO2, and pH, while discharge and SOM were negatively associated. Axis 2 was positively associated with temperature, N-NO3−, N-NH4+, and oxygen. Streams within each scenario clustered in groups in ordination space. PCA analysis showed that all three streams within each scenario remained close together during each season (dry, transition, and wet) (Fig. 2). Even during the wet season, when hydrological connectivity between the stream and its watershed is expected to be highest, we found that scenarios remained separate in ordination space (Fig. 2).

Figure 2 Principal Component Analysis of physicochemical variables at each scenario and season.

Scenarios are represented by symbols: Forest-Pasture (circles), Pasture-Forest (squares), and Forest-Coffee plantation (starts). Seasons are represented in colors and letters; dry (D, yellow), transition (T, red), and wet (W, blue). Two letters together represent the middle sampling point between land uses (e.g., PF: pasture to forest boundary).

Axis 1 ordered streams along a seasonal gradient from dry to wet season (Fig. 2). In the dry season, the three scenarios were grouped at the extreme right of axis 1 and the three sampling points (upstream, middle point, and downstream) of each scenario also grouped together. Dry season scenarios were associated with high alkalinity, pH, and BOM. The transition to the wet season clustered in the center of the graph and the wet season on the left. The wet season corresponded to the highest discharge and concentrations of SOM. Axis 2 aligned our different scenarios along a physicochemical gradient, with the P-F scenario always located in the lower quadrants of the graph, followed by the F-P scenario in the middle and the F-C scenario in the upper quadrants (Fig. 2). This gradient was consistent during the three seasons and the highest physicochemical values were those in streams draining the F-C scenario.

Effects of scenarios and seasonality

Of all variables measured (Table 2), only four presented significant differences among scenarios and seasons. Water temperature, TP, SiO2, and N-NO3− were significantly different among scenarios (Fig. 3). Water temperature and TP were significantly lower in F-P and P-F than in F-C (F = 18.3, P < 0.001; F = 11.5, P < 0.001, respectively) (Figs. 3A and 3B). SiO2 was also highest in F-C, followed by F-P, and it was lowest in P-F (F = 134.8, P < 0.001) (Fig. 3C). N-NO3− was similar between F-P and F-C, and significantly lower in P-F (H = 16.5, P < 0.001) (Fig. 3D).

Figure 3 (A–D) Comparison of major physicochemical variables among scenarios.

Only variables that were found significantly different among scenarios, without a significant scenario x season interaction, in 2-way ANOVA. Different lowercase letters within each panel indicate significant differences at P < 0.05.

We found a significant scenario-season interaction for discharge (F = 56.7, P < 0.001), conductivity (F = 9.5, P < 0.001), pH (F = 4.5, P = 0.011), N-NH4+ (F = 5.4, P = 0.005), Cl− (F = 12.2, P < 0.001), and SO42− (F = 8.8, P < 0.001). Dry season discharge was not significantly different among scenarios (Fig. 4A), but during the transition season it was significantly higher in F-P and P-F than in F-C. During the wet season, discharge was significantly higher in F-P than in P-F and F-C (Table 2). Conductivity was significantly different among scenarios and seasons (Fig. 4B), but it was consistently highest in F-C, followed by F-P, and lowest in P-F. Dry season pH was similar among scenarios (Fig. 4C), but in the transition season, it was significantly higher in F-P than in F-C; and in the wet season pH was lowest in P-F (Fig. 4C). N-NH4+ concentrations were lowest in P-F during dry and wet seasons, but were similar among scenarios in the transition season (Fig. 4D). During the dry season, Cl− was lowest in F-P and highest in F-C; this pattern changed during the other seasons, with lowest values in P-F (Fig. 4E). During the transition, Cl− was highest in F-P, but in the wet season, it was highest in F-C (Fig. 4E). SO42− had similar concentrations among scenarios in the dry and transition seasons (Fig. 4F). In the wet season, SO42− was highest in F-C and lowest in P-F.

Figure 4 (A–D) Comparison of major physicochemical variables among scenarios and seasons.

Only variables that were found significantly different in 2-way ANOVA are included. Different letters indicate significant differences at P < 0.05.

Patterns within scenarios

There were longitudinal changes in water physicochemistry, as streams flowed from one land use to another within each scenario. Water temperature showed a general increase in the downstream direction for each study stream (n = 9). In F-P, the pattern showed a positive linear trend, the lowest temperature was recorded in forest and the highest in pasture (Figs. 5A, 5D and 5G). In P-F, a similar pattern was observed, but temperature was lowest in the middle point, where the streams flowed from pasture into forest (Fig. 5B). The F-C scenario did not show strong trends, except during the transition season, when temperature increased downstream and was highest in coffee plantation (Fig. 5C). N-NO3− was generally lowest in forest than in any other land use (Figs. 6A–6I). In F-P, N-NO3− increased downstream, in pasture, except during the transition season when it was high in forest and in pasture (Figs. 6A, 6D and 6G). In P-F, N-NO3− tended to be higher in pasture than in forest, but it was always highest at the middle point, where streams flowed from pasture into forest (Figs. 6B, 6E and 6H). The F-C scenario showed a strong pattern of high N-NO3− concentrations in the coffee plantation section (Figs. 6C, 6F and 6I). TP was similar among all scenarios and seasons, with some tendency toward high concentrations in coffee plantation (Fig. 7). Chlorophyll a was consistently low in forest sections, and high values were found in pasture and coffee plantation during the dry season (Table 2).

Figure 5 (A–I) Water temperature (°C) at the three sampling points per scenario during the three sampling seasons.

Same color symbols represent each of the three studied streams per scenario. The 0 m point is uppermost section, 100 m middle point represents the boundary between the two land uses, and 200 m point is the lowermost section. Land uses are forest (F), pasture (P), and coffee plantation (C).

Figure 6 (A–I) N-NO3− concentrations (µM) at the three sampling points per scenario during the three sampling seasons.

Same color symbols represent each of the three studied streams per scenario. The 0 m point is uppermost section, 100 m middle point represents the boundary between the two land uses, and 200 m point is the lowermost section. Land uses are forest (F), pasture (P), and coffee plantation (C).

Figure 7 (A–I) TP concentrations (µM) at the three sampling points per scenario during the three sampling seasons.

Same color symbols represent each of the three studied streams per scenario. The 0 m point is uppermost section, 100 m middle point represents the boundary between the two land uses, and 200 m point is the lowermost section. Land uses are forest (F), pasture (P), and coffee plantation (C).

Stream identity played a more important role than land use in determining water physicochemistry within individual scenarios (Fig. 8). Accordingly, PCA analysis formed groups by stream number (i.e., identity) and not by land use type. In the F-P scenario, during the dry season, stream 3 had high N-NH4+, temperature, and SO42− in both forest and pasture sections (Fig. 8A). During the transition season, stream 2 differed from the rest as it had higher discharge, SO42−, and temperature (Fig. 8D). During the wet season, the pattern was similar as the one observed in the dry season (Fig. 8G). In the P-F scenario, stream 5 showed a strong physicochemical differentiation from the other two streams in all seasons, having higher N-NO3−, discharge, SiO2, alkalinity, conductivity, P-PO4, and pH than the other two (Fig. 8B). In the F-C scenario, stream 7 was also different from the other two during all seasons, having higher conductivity, alkalinity, SiO2, pH, N-NO3− and N-NH4+ than the other two streams (Figs. 8C, 8F and 8I).

Figure 8 Principal Component Analysis of physicochemical variables per each scenario and season.

Each PCA includes three sampling points at three streams within a particular scenario. (A–C) F-P, P-F, and F-C during the dry season, (D and E) the same scenarios during the transition season, and (G–I) the scenarios during the wet season.

Comparison of scenarios using Procrustes with F-P as reference indicated that both P-F and F-C scenarios had significantly different physicochemical patterns than F-P during all seasons (Table 3). Differences were of similar magnitude during the dry and transition seasons, but there was a tendency for smaller differences among scenarios during the wet season, relative to the dry and transition seasons (Table 3).

Table 3 Procrustes analysis for each scenario, using F-P as a reference for comparison.

All comparisons are significant (P < 0.01). Seasons: dry (D), transition (T), and wet (W).

Scenario	Season	Procrustes	
		m12	R	
P-F	D	0.2122	0.8876	
	T	0.1433	0.9256	
	W	0.0585	0.9703	
F-C	D	0.2424	0.8704	
	T	0.2522	0.8648	
	W	0.0949	0.9514	

Biotic responses

Mean chlorophyll a values ranged from 0.25 to 1.20 mg/m2 across scenarios (Table 2). ANOVA showed that the scenario (F = 5.36, P = 0.015) and season (F = 3.81, P = 0.42) had a significant interaction (F = 5.24, P = 0.006) over chlorophyll a concentration (Fig. 9). In the dry season, there were no significant differences in chlorophyll a concentration between scenarios (P > 0.05) (Fig. 9). However, in the transition season, concentrations in F-C scenario were significantly higher than in F-P and P-F scenarios (P < 0.05). In the wet season, P-F scenarios presented higher concentrations than F-P and F-C scenarios (Fig. 9).

Figure 9 Comparison of chlorophyll a concentrations among scenarios and seasons.

Different lowercase letters indicate significant differences in 2-way ANOVA at P < 0.05.

Few physicochemical variables were related to chlorophyll a concentrations during the dry (only Cl−) and wet seasons (pH and Ca2+) (Table 4). However, half of the physicochemical variables measured during the transition season strongly correlated with chlorophyll a (Table 4); discharge, BOM, and SOM had a significant negative correlation with chlorophyll, while temperature, alkalinity, conductivity, TP, SiO2, Na+, and K+ had a significant positive correlations (Table 4).

Table 4 Pearson correlation coefficients for the relationship between chlorophyll a and water physicochemistry.

Values in bold are significant at P < 0.05.

	Seasons	
	Dry	Transition	Wet	
Discharge	−0.304	−0.734	0.108	
Temperature	0.255	0.719	0.248	
Oxygen	0.134	0.617	−0.765	
pH	0.035	−0.082	−0.894	
Conductivity	0.316	0.894	−0.380	
TSS	0.435	−0.074	−0.031	
Alkalinity	0.400	0.922	0.120	
DIN	0.340	0.563	−0.569	
PO4	−0.022	0.370	−0.633	
TP	0.363	0.883	−0.386	
DIN/PT	0.005	−0.087	−0.381	
SiO2	0.335	0.899	−0.555	
Cl−	0.812	0.164	−0.607	
SO42−	−0.155	−0.592	−0.615	
Ca2+	−0.074	0.120	0.734	
Na+	0.292	0.917	−0.217	
Mg+	0.264	0.750	−0.314	
K+	0.362	0.872	−0.221	
BOM	0.148	−0.843	0.282	
SOM	−0.264	−0.747	−0.386	

Discussion

Our study highlights the advantage of focusing on land use combinations, or scenarios, when studying the impact of anthropogenic activities on stream ecosystems. In our case, different land use scenarios resulted in specific physicochemical characteristics in the study streams in TMCF. Those signatures were consistent over the year (during the dry, dry-wet transition, and wet seasons), indicating the driving role of land use combinations over stream characteristics. Within single scenarios (streams draining the same land use combinations), water physicochemical signatures were stream-specific, rather than related to land uses. Physicochemical patterns translated into biotic effects, as algal biomass (as chlorophyll a) was strongly related to stream characteristics in particular during the transition season.

Water physicochemistry at our study sites had the expected signature of TMCF streams. Forest streams in this area have clear waters, in particular during the dry season, with cool temperatures (<17 °C), near neutral pH (~6.5), and are solute poor (Vázquez, Aké-Castillo & Favila, 2011). Nutrients are in low concentrations. Nitrogen concentrations are in the range of 0.3–4.1 mg/L for NO3 and 0.01–1.0 mg/L for NH4. Similarly, phosphorus concentrations are low, 0–0.06 mg/L for PO4 (Vázquez, Aké-Castillo & Favila, 2011). As expected from anthropogenic impacts, most physicochemical variables increase in agricultural land uses. In pasture, most variables are higher than in forest, but nutrients tend to remain at low concentrations (NO3 < 0.3 mg/L; PO4 < 0.02 mg/L; Vázquez, Aké-Castillo & Favila, 2011). In our study, pasture sections were particularly low in most physicochemical variables (Fig. 2). Of the land uses included in our study, coffee plantation had the largest impact on stream physicochemistry, with the highest values for most variables. Most coffee plantations in TMCF are composed of shaded coffee and might have lower impacts on streams than more intense production approaches (e.g., sun coffee). However, plantations are important sources of nutrients, both N and P, and SiO2 in some cases, presumably due to fertilizer additions (Cannavo et al., 2013). A substantial area of bare soil is usual under shade coffee plantations due to weed control and high density of footpaths for coffee bean harvesting. This in turn, cause the soil to be unprotected against erosion resulting in high sediment export to streams as it has been reported in similar regions (e.g., Ramos-Scharrón & Thomaz, 2017). Although studies in cloud forest streams remain limited, the overall patterns described here agree in general with those previously described for nearby streams in the region (Vázquez, Aké-Castillo & Favila, 2011).

Land use scenarios are important drivers of water physicochemistry in TMCF streams. We documented major differences among scenarios that were consistent regardless of the season, like nutrient concentrations and water temperature. Nutrient concentrations were highest in streams draining the F-C scenario, as expected due to the intense activities in this land use that include the use of fertilizers (Vázquez, Aké-Castillo & Favila, 2011). The P-F scenario had relatively low values of stream physicochemistry, as reflected by its position in ordination space (e.g., Fig. 2) and in agreement with previous studies (Vázquez, Aké-Castillo & Favila, 2011). In the case of water temperature, it was lowest in streams that started in forest and drained into pasture (i.e., F-P) and highest in scenarios that included coffee plantation. However, streams draining coffee plantation were located at slightly lower elevations (1,070–1,420 m a.s.l.) and their warmer temperatures could be expected, as other scenarios were all above 1,400 m a.s.l. (Table 1).

Other physicochemical differences among scenarios had a seasonal behavior, increasing or decreasing in different times of the year. The studied streams drain volcanic soils and underlying semi-permeable fractured andesitic-basaltic bedrock, which confers the watersheds both high permeability and high water retention capacity (Solano-Rivera et al., 2019). As such, the physicochemistry signature observed during the dry season reflects subsurface waters and the associated lithology (Muñoz-Villers & McDonnell, 2012, 2013) as the watershed is less hydrologically connected (i.e., less influence of adjacent slope or riparian land use). During the transition from dry to wet seasons, the observed increases in sediment and nutrient loads could be associated with the occurrence of intense rainfall events when the soil mostly dry. Lastly, with increasing antecedent wetness and hydrological connectivity of the watersheds during the wet season, differences among scenarios could be expected to decrease (Muñoz-Villers & McDonnell, 2013). Interestingly, while seasonality changed the physicochemical signature of our study scenarios, it did not mask the effect of particular land uses over the stream. Stream water conductivity for example, was always higher in scenarios that included coffee plantation, supporting the larger impact of the land management involved in the production of shaded coffee over stream sediment and nutrient loads regardless of seasonality. Other variables showed similar patterns in each season, but their concentrations were highest during the wet season (e.g., Cl, SO4). Overall, seasonality played a role determining the physicochemical characteristics of streams draining different scenarios. However, those effects did not change the general patterns created by the combination of land used within each scenario.

A close up examination of streams within each scenario highlighted the important role that forest sections played in buffering and even restoring water physicochemical characteristics. As expected, stream sections draining forest had low physicochemical values (e.g., temperature and nutrient concentrations); the lowest values were often found at our middle sampling point that represented the downstream point of the forest section (i.e., the 100 m point). Variables increased as streams drained pasture or shaded coffee plantation, reflecting anthropogenic land use impacts. In the P-F scenario during the dry and transition seasons, we clearly observed some degree of recovery, with nutrients decreasing in the downstream forest section (Figs. 6B, 6E and 6H). Our findings support the observation that forest remnants affect water characteristics and that even small-scale heterogeneity within remnants may influence their role as buffers (de F Fernandes, de Souza & Tanaka, 2014). In addition, it highlights the importance of conservation and restoration of riparian vegetation along streams to improve stream water quality. In agricultural areas, heterogeneous landscapes might protect stream ecosystems by providing patches of forest were streams recover (at least partially) from upstream impacts (e.g., de Paula et al., 2018).

Stream identity had a stronger role than land use type in determining water physicochemistry when comparing the effects of both under individual scenarios. The three selected streams within each scenario were expected to behave like replicates and sections draining particular land uses to share similar physicochemical characteristics. While land use clearly affected stream physicochemistry, our sampling points grouped in ordination space based on stream identity rather than land use. This more important role of stream identity might be related to the high spatial variability in topography and geoforms within each watershed inherent to the mountainous landscape (Pike, Scatena & Wohl, 2010). In addition, differences in watershed shape, drainage area and stream morphology, which in turn cause a wide variation of soil properties, water flow paths (both surface and subsurface), residence time, and discharge magnitude, might ultimately result in particular stream water physicochemical signatures (Engelhardt, Weisberg & Chambers, 2012; Bleich et al., 2014).

The effects of water physicochemistry generated by each scenario had a clear impact on stream ecosystem function, as indicated by algal biomass. Chlorophyll a concentration was our proxy to understand whether scenarios resulted in changes in ecosystem components. Benthic algal biomass depends on variables like land use, a large-scale factor, and water velocity, solute composition, and nutrients, which are local-scale factors (Urrea-Clos, García-Berthou & Sabater, 2014). The conditions that favor high chlorophyll concentrations are steady hydrological conditions and shallow waters that receive high light irradiances (Urrea-Clos, García-Berthou & Sabater, 2014). In our study, scenarios that included high solar radiations (i.e., pasture) and high nutrients (i.e., shaded coffee plantation) also had the highest benthic algal biomass during the dry season. The lack of a strong algal response on the other seasons could be due to hydrological disturbances.

Chlorophyll a values at >100 mg/m2 are considered indicators of high nutrient concentrations (TP and TN). Chlorophyll a concentration in our study streams ranged from 0.25 to 1.2 mg/m2, that is, low concentrations. These results could be explained by the fact that our study streams are small and headwater (1st and 2nd order), heavily shaded, and there is no urban influence. The highest values were recorded in streams in the F-C scenario (average 0.90 mg/m2). Those streams had the highest temperatures and availability of nutrients during all seasons that were potentially driving the chlorophyll a values. TP and water temperature are two important drivers of algal biomass (Munn, Waite & Konrad, 2018). Similar results have been reported in many streams in agricultural areas with the highest concentrations of nutrients (Munn, Frey & Tesoriero, 2010; Vázquez, Aké-Castillo & Favila, 2011; Urrea-Clos, García-Berthou & Sabater, 2014).

Conclusions

Our study highlights the value of assessing upstream-downstream combinations of land uses, as scenarios, rather than contrasting single land uses when studying land use effect on stream ecosystems. Landscapes in TMCF are particularly fragmented and stream ecosystems drain a mosaic of land uses. We found that those combinations of land use generate particular stream characteristics that are consistent over the year and not explained by a single land use alone. We also found evidence indicating that stream identity (e.g., combination of topographical and geomorphological aspects) plays a key role explaining water physicochemistry, instead of only land use. Our findings suggest that stream location (e.g., elevation) are at least as important as land use in defining water physicochemistry. Finally, our longitudinal data provides support for the role of forest fragments in buffering anthropogenic land use impacts on stream ecosystems, as well as for the protection and restoration of forested riparian vegetation. Similar to our study, others have stressed how forest fragments play a key role protecting fragmented tropical landscapes (Arce-Peña et al., 2022). Given that most tropical streams drain heterogeneous landscapes, the presence of patches of forest could be crucial in protecting stream ecosystems from anthropogenic land use impacts.

We appreciate help provided by Javier Tolome and Víctor Vasquez, in the field; and Ariadna Martínez and Daniela Cela for support with chemical analysis.

Additional Information and Declarations

Competing Interests

Author Contributions

Data Availability

The authors declare that they have no competing interests.

Gabriela Vázquez conceived and designed the experiments, performed the experiments, analyzed the data, prepared figures and/or tables, authored or reviewed drafts of the article, and approved the final draft.

Alonso Ramírez conceived and designed the experiments, performed the experiments, analyzed the data, prepared figures and/or tables, authored or reviewed drafts of the article, and approved the final draft.

Mario E Favila analyzed the data, prepared figures and/or tables, authored or reviewed drafts of the article, and approved the final draft.

M Susana Alvarado-Barrientos conceived and designed the experiments, performed the experiments, analyzed the data, authored or reviewed drafts of the article, and approved the final draft.

The following information was supplied regarding data availability:

The data is available at Zenodo: Ramírez, Alonso & Vázquez, Gabriela. (2022). Stream Physicochemistry, La Antigua Watershed [Data set]. Zenodo. https://doi.org/10.5281/zenodo.7083510.

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
