# Peer review of "Land use scenarios, seasonality, and stream identity determine the water physicochemistry of tropical cloud forest streams"

_PeerJ, doi:10.7717/peerj.15487_

## Round 0.1 · original submission · Major Revisions

While all three reviews agree that your research questions are relevant and that your study provides an interesting dataset, two of the revisers have point to a number of shortcomings.

Generally speaking, the study methods and the results are not explains in a convincing way, some data (water chemistry) is missing as well as important details are missing while other parts are explained several times. The discussion part needs to be more clear and focused. Reviewer #3 suggests that most of the manuscript be reorganized and -written and that a synthesis should be added.

Please consider and address all comments made by all three reviewers. Explain carefully how you decide to handle their suggestions. Please note that reviewer #3 has left an annotated version of the manuscript.

I hope you find the suggestions usefull.

Reviewer 1 ·

Basic reporting

The authors of Land use scenarios, seasonality, and stream identity determine water physicochemistry in tropical cloud forest streams have a presented a nice study describing the physico-chemical patterns in streams draining different scenarios of land-use in a tropical forest. They show that land-use has a direct influence on stream chemistry, conductivity, temperature, pH and but seasonality also has direct impacts on these characteristics and patterns. The paper is generally nicely and clearly written, although I have some very minor comments below to help improve the readability of the results sections that is a bit dense. All other parts of the paper are very well written.

Experimental design

The experimental design is also well put together and addresses interesting questions within tropical stream biogeochemistry. The authors have collected a large database of different analyses cross 9 different streams and presented the data in numerous statistical analyses to disentangle the role of land0use vs seasons but also difference in streams across land-use and seasons.

Validity of the findings

Conclusion sections is very well done and clear. They summarize the major pints of theur study and are supported by the literature.

Additional comments

General Comments:

Line 39: should be chloride and sulfate, these should not be plural (same as in line 163 and 164)

Line 106-107: The mention of volcanic lithology seems a bit out of place; either remove it or elaborate a bit more on the role of lithology

Line 153: Perhaps instead use F-P, P-F, and F-C, as a reminder of what is meant by the scenarios. This might be more direct than "each stream draining two land use"

Line 163-164: Si, Cl and So4 are missing their respective methods

Line 242: The SOM range is very large, a quick mention of what scenario had the highest, as you did in the previous sentence

Line 250-251: This is really cool ( and the seasonal gradient below) and that it comes out so clearly in the PCA is really nice!

Line 267-297: This section is a lot of details, perhaps you can add a 1-2 summary sentences at the end something along the lines of, Conductivity, Cl, NH4, and SO4 showed the greatest seasons variation, especially for streams in F-C that consistently exhibited the highest concentrations. Just as an example, adapt it to how you see fit.

Also, in the methods you mention many more parameters than these, is there a specific reason for highlighting these parameters in figure 3? I see the answer to this is in the Fig 3 caption. Maybe add a sentence in the results listing the parameters that shows no statistical differences, this is also a result.

Line 300-354: A comment similar to the previous one for this section. Again it is a lot of details per scenario and season. Consider adding a summarized take home message to each paraph or figure/results description to help the reader digest a bit more all the details of the results.

Line 401: add to after due

Line 420: Was elevation included in your PCAs? This might be another important variable in additions to land use, although it’s kind of related.

Line 441-443: This a really nice summary of your findings.

Line 492: Should PT be TP?

Figures 4-6: The different land-uses are not very clearly distinguished in these three figures. Would it be possible to add a legend? This would help interpret the different colors and shapes, although the pints are rather small and hard to distinguish.

·

Basic reporting

Water chemistry raw data is not shared. Table 2 provides average per type of stream but not for each of them. The authors can easily fix that by providing a table with all the data in the appendix or submitting the data on an open repository (e.g. https://edirepository.org/) and refer it on their paper

Experimental design

The research question is well defined and relevant. The authors provided an interesting dataset for streams in tropical cloud forest in Mexico, that link the type of riparian vegetation (100m strip from the stream) to the water chemistry and biological variables (chl a in rocks and BOM) for three sampling periods.
The design of the study takes into account three possible scenarios, that are composed by a combination of land use in the riparian area (Forest-Pasture, Pasture- Forest, Forest-coffee plantations) and sampled in three stations (up, mid and down).
I think this is interesting to consider the mosaic configuration; however, I find this is not well resolved for some of the analyses:
- Not stated how the subwatershed draining the streams might influence water chemistry (only a 100m buffer riparian area selected). This is especially important for the UP station, which the water chemistry should depend on the watershed upstream but also for MID and DOWN stations on where the influence on water chemistry might be dependent from upstream water.
For example, I think Fig. 1, 2, and 3 includes UP station, which might not be affected by the surrounding riparian area (but upstream land use). A possibility to resolve this would be to use similar approaches previously used (Shrogen et al. 2019 and Abbott et al. 2018) accounting upstream station chemistry.
- I would recommend to include a map with the land use of watershed and subwatershed of the area to be able to understand land uses in the whole watershed.
- I would also find interesting to understand why the authors select 100m buffer riparian strip. Would be possible to discuss the effect of a wider riparian buffer (e.g. doing a sensitivity analyses on the effect of the riparian buffer selecting other widths)?
Some methods are not clearly explained. For example canopy cover and slope riparian zone (in Table 1) is not explained in the methods. The explanation for other methods could also be more detailed, e.g. filtration of the water samples and storage or average dimension of the rock samples (were big or small?) how the rock surface area was measured. Also, the number of stream per each scenario is not stated in the methods.

Validity of the findings

No comment

Additional comments

No comment

Reviewer 3 ·

Basic reporting

This article studied the physicochemical variability of 3 headwater streams representing different land use patterns (Forest-Pasture; Pasture-Forest and Forest-Coffee). The authors explained their findings with respect to seasonality as well. This is an interesting research which has generated good information on a relevant subject. Having said that, the MS needs to be much more concise to improve clarity and readability.

The Introduction needs improvement, mostly in the organization and sequence of the ideas. The authors need to avoid repetition and structure the writing from the general to the specifics, also stating what is already known regarding effects of land use in the P-C characteristics of mid elevation streams. At this moment, it lacks appropriate threading of the ideas, as they seem isolated from one another. Details can be seen below as comments on the original MS (attached).

Figures have good quality and English language is grammatically correct.

Experimental design

The methodology section requires improvement, some details are missing or are not clear.

Some parts of the Introduction or the Discussion belong in the methodology section.

Comments can be seen below on the original MS (attached).

Validity of the findings

Some of the results have to be re-checked for accuracy in the interpretation.
All underlying data have been provided, but the writing makes the findings difficult to follow.

Also, the Discussion needs to be reorganized, I suggest to divide it into subsections, because the writing is diffuse and jumps from one part of the research to another, decreasing readability and comprehension of ideas.

Conclusions should address only what was shown with the results generated in this investigation.

Additional comments

As stated before, I find this research to be interesting and valuable for publication, however, the article needs to be much clearer, better written. Seems to me that a rigorous understanding of the background knowledge on the field is missing, and that a more in-depth interpretation of the results is needed. Also several results are repeated in text, Tables and figures (for example Table 2 and Figures 2 and 3).
Figures 4,5,6 should be in the Supplementary Information. I believe a summary of these figures is seen with Figure 1 or Figure 7. The authors must choose which figures or tables render the most information and avoid repeating the results in different ways.

As it is, the results section is very hard to follow and becomes confusing. It also has too much text and lacks an appropriate synthesis. I suggest re-structuring a lot of the text (principally the “Effects of scenarios and seasonality” and the “Patterns within scenarios” sections) and condense it as a Table or other visual tool. The reader cannot process the results as they are shown.
Same thing happens with the discussion, which need re-organization.

Annotated reviews are not available for download in order to protect the identity of reviewers who chose to remain anonymous.

---

## Round 0.2 · Minor Revisions

Your manuscript has been improved considerably from incorporation of the the comments from the reviewers. One of the previous reviewers has evaluated the revised version and is satisfied with the new version. So am I but want to address two things. The first is that I agree with your decision not to include elevation in the PCA due to co-correlation. The other thing is if you have decided to include the raw data or not. In you comment to Reviewer 2, who suggest to place the raw data in a repository, that these data can be obtained from the author, while you in your reposes to Reviewer 3 (R30) says: ´The raw data is now deposited in Zenodo.org and accessible to readers´. Please verify that the latter has happened and provide a link in the manuscript to where the data can be found.

Reviewer 1 ·

Basic reporting

The writing is clear and professional, the use of references and background context is appropriate, and the figures and tables are all clear.

Experimental design

The research questions and objectives are clearly stated and the methods answer them very well. Their study helps to better understand land-use changed in tropical regions and how they influence stream chemistry which is very appropriate and needed.

Validity of the findings

The authors have improved the clarity of the methods and results sections and manuscript as a whole flows much better now.

---

## Round 0.3 · accepted · Accept

Thank you for clarifying the supplementary data. I´m happy to see all your work has been fruitful.

---

## Author Rebuttal · Round 0.3

April 8, 2023

Kirsten Christoffersen
Academic Editor
Dear Dr. Christoffersen,

We completed our revision of the manuscript (#2022:09:77297:1:1:REVIEW - Land use scenarios, seasonality, and stream identity determine the water physicochemistry of tropical cloud forest streams). We appreciate your time and that of the reviewers in helping us improve this manuscript.

Addressing your comments:
**"The first is that I agree with your decision not to include elevation in the PCA due to co-correlation."**
No action taken.

**"The other thing is if you have decided to include the raw data or not. In you comment to Reviewer 2, who suggest to place the raw data in a repository, that these data can be obtained from the author, while you in your reposes to Reviewer 3 (R30) says: ´The raw data is now deposited in Zenodo.org and accessible to readers´. Please verify that the latter has happened and provide a link in the manuscript to where the data can be found."**
We apologize for the mixed messages. The raw data is available in Zenodo, and the detailed information is under "Associated Data." The text reads:

Data is available at Zenodo as: Ramírez, Alonso & Vázquez, Gabriela. (2022). Stream Physicochemistry, La Antigua Watershed [Data set]. Zenodo. https://doi.org/10.5281/zenodo.7083510

I understand that this statement is automatically added to the publication, so I did not add this to the manuscript.  If this is incorrect, I will be happy to add a statement in the methods.

We will be happy to provide further information if necessary.

Sincerely,

**ALONSO RAMÍREZ**
Professor